# Differences in Anthropometric and Body Composition Factors of Blind 5-a-Side Soccer Players in Response to Playing Position: A Systematic Review

**DOI:** 10.3390/jfmk10030238

**Published:** 2025-06-24

**Authors:** Boryi A. Becerra-Patiño, Armando Monterrosa-Quintero, Jorge Olivares-Arancibia, José Francisco López-Gil, José Pino-Ortega

**Affiliations:** 1Programa de Doctorado en Ciencias de la Actividad Física y del Deporte, University of Murcia, San Javier, 30720 Murcia, Spain; boryialexander.becerrap@um.es; 2Facultad de Educación Física, Universidad Pedagógica Nacional, Bogotá 110221, Colombia; 3Altius Performance Laboratory, Physical Education and Sports Program, Universidad Surcolombiana, Neiva 410001, Colombia; adomonterrosa@gmail.com; 4AFySE Group, Research in Physical Activity and School Health, School of Physical Education, Faculty of Education, Universidad de las Américas, Santiago 7500975, Chile; jorge.olivares.ar@gmail.com; 5School of Medicine, Universidad Espíritu Santo, Samborondón 092301, Ecuador; 6Vicerrectoría de Investigación y Posgrado, Universidad de los Lagos, Osorno 5290000, Chile; 7Faculty of Sport Science, University of Murcia, 30100 Murcia, Spain; josepinoortega@um.es

**Keywords:** blind soccer, body composition, anthropometry, somatotype, body weight

## Abstract

**Background**: Blind 5-a-side soccer is an intermittent sport that requires the integration of physiological and physical processes, where body composition (BC) is an influential and differentiating factor of the sporting level, according to the conclusions of some studies. However, to date, no systematic review has been reported comparing BC in players with visual impairment. **Objectives:** The aims of this study were to systematically synthesize the existing evidence on differences in anthropometric characteristics and body composition among blind 5-a-side football players according to playing position and to derive practical recommendations for researchers and coaches. **Methods:** The following databases were consulted: PubMed (Medline), Scopus, Web of Science, and Science. This systematic review uses the guidelines of the PRISMA declaration and the guidelines for conducting systematic reviews in sports science. PICO strategy was used for the selection and inclusion of studies in the present work, with a series of inclusion and exclusion criteria. The quality was methodologically assessed using the PEDro scale. **Results**: The 10 studies comprising this systematic review had a total sample size of 168 athletes. The main findings of this research were (1) the somatotype of blind 5-a-side soccer players tends toward meso-endomorphic; (2) there are differences in the variables of muscle mass, fat mass, and body weight in response to playing position and sporting level; (3) the players present a somatotypical profile with a predominance of the mesomorph component. **Conclusions**: The results of this review reveal a tendency to define BW as influencing the athletic performance of blind 5-a-side soccer players. However, it is not conclusive whether these improvements occur in response to each playing position. More studies are needed to analyze the effect of BW on athletic performance, especially when correlating BW with other physical, nutritional, technical, and tactical variables in training and competition.

## 1. Introduction

The study of body composition (BC) in sports aims to evaluate the body’s reserves through different variables [1], including fat-free mass (FFM), fat mass (FM), and muscle mass (MM), to assess an individual’s nutritional status and promote nutritional processes [2,3]. BC allows the determination of values that are important for athletic performance and athlete health [4,5,6], such as the amount of skeletal muscle, body density, MM, metabolic balance, muscle-fat ratio, skeletal index, and somatotype, among others [7,8].

Athletic performance is a multifactorial, complex, and dynamic process [9,10,11] that requires the integration of various physical factors, including strength to perform specific actions of acceleration, deceleration, changes in direction, and jumping [12,13,14]; speed to cover distances in short times [15,16]; endurance to maintain the quality of actions performed at high intensity levels [17,18]; and flexibility to improve the quality of actions performed [19,20]. Likewise, relevant factors at the physiological level highlight the processes of adaptation, recovery, energy systems, and BC [10,21]. Therefore, investigating the influence of these factors on sports is crucial for promoting increasingly specific training processes [22,23].

In this sense, BC has been established as a relevant and determining factor in athletic performance in team sports [24,25], especially in women’s soccer [26,27], men’s soccer [28,29], soccer players with cerebral palsy [30,31], and, consequently, blind soccer players [32,33,34,35,36,37,38,39,40,41]. Each sport has specific characteristics, requiring athletes to adapt to competitive demands, many of which are imposed by a series of actions that they must perform in response to their playing position [35,36,42]. Thus, blind 5-a-side is a physically, physiologically, technically, and tactically demanding sport [32,42,43,44]. Blind 5-a-side is an intermittent sport that combines the aerobic-anaerobic system, seeking to demonstrate optimal cardiovascular endurance, agility to quickly change direction and orientation [45], speed for dribbling, strength and power for shooting, acceleration, deceleration, and coordination for passing and positioning [46,47,48]. Thus, blind 5-a-side being a sport that promotes intense efforts with short recovery phases requires athletes to have high levels of MM, low levels of body fat percentage, and a mesomorphic somatotype profile [33,35,37,38,40]. Some studies have analyzed the BC of blind 5-a-side players, determining that each playing position has specific requirements and that these requirements are related to physical abilities [37,40]. However, other studies have reported that, although there is homogeneity in anthropometric and BC factors, the different playing positions do not differ [35,41]. Another determining variable when analyzing whether different playing positions express variations in BC is the inclusion of goalkeepers in the studies. Thus, the study by Gorla et al. [34], which included goalkeepers in the evaluation, determined significant differences compared with other positions.

To date, no systematic review has compiled the available evidence on the differences in anthropometric factors, BC characteristics, and somatotypic profiles of blind 5-a-side players, and little research has been conducted on this sport. Therefore, the aims of this study were to systematically synthesize the existing evidence on differences in anthropometric characteristics and body composition among blind 5-a-side football players according to playing position and to derive practical recommendations for researchers and coaches.

## 2. Materials and Methods

### 2.1. Design

This systematic review followed the guidelines of the Preferred Reporting Items for Systematic Reviews and Meta-Analyses (PRISMA) statement [49,50] and the sports science guidelines for conducting systematic reviews [51]. The review protocol was registered in International Platform of Registered Systematic Review and Meta-analysis Protocols (INPLASY) website on 18 June 2025 (ID 202560075).

### 2.2. Sources of Information

The search strategies considered the following characteristics. Date: All studies published up to 15 March 2025 were retrieved. The following databases were consulted: PubMed (Medline), Scopus, Web of Science, Science Direct, and SPORTDiscus. Google Scholar and ResearchGate were also searched. These databases were consulted for use in various reviews and were used to search databases and other sources.

### 2.3. Inclusion and Exclusion Criteria

Two authors searched independently (B.A.B.-P. and J.O.-A.). The purpose was to identify papers that met the criteria (Table 1). After the selected studies were identified, the comma-separated value (CSV) file was downloaded, and relevant criteria for study selection were defined (title, keywords, abstract, year, journal, citations received). Documents were screened to remove duplicates. Furthermore, if any documents were found and not captured by the search equation, they were added through external sources. For the selection and inclusion of studies in this study, a series of inclusion and exclusion criteria were established on the basis of the participants, interventions, comparison, and outcomes (PICO) strategy (Table 1). Any document that included a comparison between blind and sighted players within the research area was excluded. The inclusion criteria were as follows: (i) studies published without language restrictions and (ii) original studies. The exclusion criteria were as follows: (i) systematic reviews, meta-analyses, bibliometric analyses, narrative or literary reviews; (ii) abstracts, meetings, books, reviews, letters, and editorials; (iii) articles written without academic peer review; and (iv) studies without full access to the original text.

### 2.4. Search Strategy and Data Extraction

The 10 studies included in this systematic review included a total sample of 168 athletes. To design the search strategy, the P (population), I (intervention), C (comparison), and O (outcomes) strategies were applied, as suggested by the guidelines used for this systematic review [52]. The Boolean operators “AND” and “OR” were used to group the terms. A similar procedure was followed for each database. Before the final search phrase for each database was constructed, possible combinations were tested with the following list of words: (“Athletes of 5-a-side Football” [All fields]) OR (“blind soccer” [All fields]) OR (“FA5 for blind persons” [All fields]) OR (“w5-a-side football team Paralympic” [All fields]) OR (“5-a-side football team” [All fields]) AND (“body composition” OR somatotype OR anthropometry [All fields]). From these terms, the following search equation was constructed: (“Athletes of 5-a-side Football” OR “blind soccer” OR “FA5 for blind persons” OR “w5-a-side football team Paralympic” OR “5-a-side football team”) AND (“body composition” OR somatotype OR anthropometry). This search string was adapted for the databases and the other methods. The controlled vocabulary search was performed with the keyword search to improve retrieval. Searches were conducted to identify studies without other restrictions regarding publication date, language, or study design. Citation searches were also performed for key included studies, with the goal of tracking other documents. When it was not possible to obtain the full texts of articles from institutional or open access subscriptions, attempts were made to contact the corresponding authors directly through the ResearchGate platform. Furthermore, if a document was found that did not appear in the search strategy, it was added through external sources.

All the retrieved articles were analyzed for duplicate entries. Two authors (B.A.B.-P. and J.O.-A.) independently reviewed the different searches to determine the terms that yielded the greatest number of documents related to the topic. Any disagreement (5% of the total documents) regarding the final inclusion/exclusion status was resolved through academic discussion, both in the selection and inclusion phases. During the discussion, the two independent authors simultaneously analyzed the articles following the criteria established in the order shown in Table 2. This process was systematized in Excel. The academic debates for the inclusion of the studies took into account the duplicate search by two authors on two different days to review the documents. In particular, the methodology (study design, variables, instruments, determination of fat percentage, and somatotype) was reviewed, as well as the results and main conclusions.

## 3. Results

### 3.1. Identification and Selection of Studies

A total of 75 documents were identified. After an initial review of the final database, documents were eliminated because of duplication (*n* = 10), leaving four for the databases and six for the other methods. Documents that were not related to the topic after the title/abstract/keywords had been reviewed were excluded from the databases (*n* = 62) or other methods (*n* = 137). A total of 199 studies were excluded. Thirteen screened documents were analyzed in depth through a systematic reading (Figure 1). After this analysis, 10 studies met the eligibility criteria. Table 3 was compiled to contextualize the sample for each of the included studies.

### 3.2. Methodological Quality

The methodological quality of the articles included in this review was assessed via the PEDro scale [53]. This scale is based on criteria that allow the identification of whether the studies have sufficient internal validity and statistical information to interpret the results (external validity (item 1), internal validity (items 2–9), and statistical information (items 10–11). Each item was classified as yes or no (1 or 0, respectively), depending on whether the criterion was met in the study. The total score considers items 2 to 11; therefore, the maximum score was 8 [32]. Regarding the quality of the evidence, scores < 4 are considered poor quality, scores ranging from 4–5 moderate quality, scores ranging from 6–8 good, and scores ranging from 9–10 excellent [41]. In this review, 100 items (97.5%) were assessed by agreement between two reviewers, and the remaining items were assessed according to the mean of the studies (Table 2). The methodological quality ranged from “moderate to good” since some studies did not present randomization in the selection of the sample, nor did they have a control group. Furthermore, the methodological quality was heterogeneous across all studies. Therefore, the methodological quality was defined by the consensus of the investigators as “moderate”, indicating differences in the methodological rigor of the included studies [53].

### 3.3. Analysis of the Participants

The 10 studies comprising the sample of this systematic review included 168 athletes, all of whom were men. Table 3 specifies the characteristics of the sample selected.

### 3.4. Analysis of the Studies

Table 4 specifies the characteristics of the sample selected for each study (study aim, variables, results, instruments, determination of % fat and somatotype, conclusions).

## 4. Discussion

To our knowledge, this is the first systematic review to analyze BC in blind 5-a-side footballers both in general and in response to playing position. The main findings of this study were as follows: (1) the somatotype of blind 5-a-side football players tends toward meso-endomorphic [35,36,38,40]; (2) there are differences in MM, FM, and BW variables in response to playing position and sporting level; (3) the players present a somatotypic profile with a predominance of the muscular component; (4) different formulas are used in the studies, although the most common are the Siri formula to determine body fat percentage on the basis of body density from other equations, the Jackson & Pollock [57] equation for BD, and the Heath–Carter method for somatotyping [55]; and (5) no significant differences were observed in the absolute values of body mass, BC, and somatotype after 16 weeks of training.

### 4.1. Body Composition and Anthropometric Factors in Blind 5-a-Side Football Players

In blind 5-a-side football, different playing positions require specific demands [60,61,62]. However, most studies on blind 5-a-side football have analyzed BC in cross-sectional studies [33,34,35,36,37,38,39,40,41], and only one longitudinal study has been reported, concluding that 16 weeks of training did not produce significant changes in the players’ BC or somatotype [32].

Among the anthropometric results, it is noteworthy that blind 5-a-side football players have a body fat percentage ranging from 10.4% to 15.9%, and the sum of the nine skinfold measurements reveals a percentage ranging from 89.7% to 121.8% in blind Brazilian national team football players [34]. On the other hand, the study conducted by Durán-Agüero et al. [33] revealed that Chilean elite Paralympic 5-a-side football players have a body fat percentage of 25.8%, an MM of 45.6%, and a bone mass of 12.1%. Another study reported that the body fat percentage was 16.23%, FM was 11.08 kg, and FFM was 57.79 kg [39]. Another study reported that the body fat mass of blind 5-a-side football players is 11.5 ± 2.7 kg, the bone mass is 11.7 ± 1.1 kg, the bone mass percentage is 16.3 ± 1.2%, and the body fat percentage is 15.9 ± 2.9 [35]. Other studies reported that blind 5-a-side footballers had body fat percentage values of 15.9 ± 2.9 mm, and the sum of the nine skinfolds was 76.9 ± 17.1 mm [36]. A study that analyzed the body components of blind 5-a-side players at the national level reported higher body fat percentage values (20.4 ± 5.1 mm), a sum of the nine skinfolds of 119.4 ± 5 mm, and a bone mass percentage similar to that reported in another study (16.1 ± 0.9 mm). On the other hand, the MM percentage was low (39.5 ± 3.5 mm) [37].

A study conducted by Gorla et al. [34] revealed that goalkeepers had the highest body weight (82.3 kg), followed by pivots (71.4 kg), defenders (70.8 kg), and wings (68.5 kg). The BF of goalkeepers was higher (21.5%) than that of wings, who presented the lowest BF (10.6%). In another context, with blind 5-a-side football players of the Spanish national team, the FM was 12.55 ± 6.21 kg and the FFM was 61.34 ± 6.36 kg [41].

### 4.2. Somatotype Values in Response to Playing Position

With respect to somatotype, there is a clear tendency for blind 5-a-side footballers to exhibit a meso-endomorphic profile. However, these approaches also reflect that, methodologically, some studies analyze somatotype generally, whereas others specify it by playing position. In the study conducted by Durán-Agüero et al. [33], there was a tendency toward a mesomorphic somatotype profile, followed by endomorphism in elite Chilean players.

With respect to playing positions, the study by Gorla et al. [34] revealed that goalkeepers (*n* = 4) tended to endomorph–mesomorph, wing (*n* = 7) toward endo-mesomorph, defender (*n* = 6) toward balanced mesomorph, pivot (*n* = 6) toward endo-mesomorph, and at a general level (*n* = 23) toward endo-mesomorph. In this same study, no significant differences were found (*p* ≤ 0.05) between the components in terms of ectomorphy, mesomorphy, and endomorphy related to playing position. When stratifying them by playing position, goalkeepers present an endo-mesomorphic profile, whereas defenders present a balanced mesomorphic profile [34]. A study analyzing the dermatoglyphic profile and BC in blind Brazilian national football team players revealed that goalkeepers tend toward meso-endomorphic characteristics, fix and pivot players toward balanced mesomorphism, and, finally, wing players toward meso-endomorphic characteristics [36].

A study analyzing the somatotype frequency distribution in response to playing position in blind football players revealed that goalkeepers have a 100% meso-endomorphic distribution, with 66.6% balanced mesomorphic characteristics and 33.3% meso-endomorphic characteristics, wing players with 71.5% meso-endomorphic characteristics and 28.5% balanced mesomorphic characteristics, and pivot players with 100% balanced mesomorphic characteristics [35]. Moreover, at a general level, the distribution of the 15 players evaluated was 60% meso-endomorphic and 40% balanced mesomorphic [35]. Moreover, a recent study with players (*n* = 63) from various high-performance 5-a-side football teams revealed that 69.2% of pivot players (*n* = 13) presented with meso-endomorphs, 15.4% with balanced mesomorphs, and 15.4% ectomorph–mesomorph. The wing players (*n* = 24) presented with 58.3% mesomorphs, 20.6% with endomorphs and mesomorphs, 15.9% with balanced mesomorphs, 4.8% with ecto-mesomorphs, 3.2 with endo-mesomorphs, 1.6% with meso-ecotomorphs, and 1.6% with ectomorphs [40].

### 4.3. Influence of BC on the Athletic Performance of Blind 5-a-Side Football Players

The skeletal index was correlated with the ball-handling speed (*r* = 0.85, *p* = 0.01). It has a moderate correlation with lower limb length (*r* = 0.69), as well as with other anthropometric variables in which muscle tissue stands out: mesomorphism (*r* = 0.59), MM (*r* = 0.57), thigh muscle area (*r* = 0.56), and calf muscle area (*r* = 0.55) [38]. Similarly, there is a relationship between dermatoglyphic characteristics and BC in blind 5-a-side football players, as evidenced by a somatotype profile with a predominance of the muscular component, which is related to the genetic predisposition toward the development of speed and strength [36]. Among the roles played by blind 5-a-side football players, laterality has been reported to be related to BC, with ambidextrous players showing lower values than other players (left- or right-handed) [41].

In this regard, and following other systematic reviews on BC analysis in football, the use of different measurement protocols can lead to significant differences in the data reported for the groups (*p* < 0.001) [6]. This is similar to the case reported in the present study, where different anthropometric formulas, equipment, and indicators are considered in each study in a heterogeneous manner. Moreover, the performance of anthropometric measurements under protocols established by international organizations such as those of the International Society for the Advancement of Kineanthropometry (ISAK) is not evident in the manuscripts studied [63,64].

Furthermore, when the playing position was used as a reference, it was determined that there were significant differences in response to BW, the sum of the skinfolds, the MM (kg), and the FFM (*p* = 0.001). No differences were reported in the percentages of MM, FM, or bone mass or in the somatotype [29]. These results are in line with those reported in the present review with blind 5-a-side futsal players. Finally, what was reported in futsal players with cerebral palsy reveals that the anthropometric and BC profiles do not vary with the functional classification, and the expressed somatotypic profile is meso-endomorphic, a case similar to that reported in futsal players with visual impairment, where there is also a tendency toward endo-mesomorphic [65]. These findings reveal the need to continue researching the effects of BC on futsal players with disabilities.

#### 4.3.1. Limitations and Strengths

This study has several limitations, which are outlined below. There is diversity in the number of participants evaluated: the most commonly used study designs have been cross-sectional, with only one longitudinal study reported. Another limitation is that not all studies include goalkeepers, making it difficult to generate a deep understanding of BC in response to playing positions. Similarly, there is diversity in the variables that determine anthropometric and BC factors. This prevented comparative measurements for each of the variables analyzed in the studies included in this review. The studies included in this systematic review were too heterogeneous and of moderate methodological quality, making it impossible to conduct a meta-analysis. Although this type of study, which was conducted to analyze blind 5-a-side football, does not allow for solid conclusions, the information contained in Table 4 reflects important information from each study that could be further explored by the scientific community.

#### 4.3.2. Future Recommendations

Future research directions for studying BC in blind 5-a-side football should seek associations with other performance indicators, especially nutritional, physical, and external load variables in competitions. Likewise, longitudinal studies, randomized controlled trials, or principal component analyses based on standardized and recognized protocols are needed to understand how different variables are related to playing position. Furthermore, the findings of this study should be carefully analyzed to be incorporated into different training processes and the practical work of coaches and athletes. Finally, it would be important to conduct BC assessments in blind female 5-a-side football players, as no studies analyzing these factors in this population sample have been reported. This will allow us to further expand the horizons of this sport because it is still a practice that has produced low scientific productivity and requires an increasing amount of research [66].

#### 4.3.3. Practical Applications

A particularly noteworthy result is the lack of significant somatotype differentiation across player positions, with a general dominance of the mesomorphic–endomorphic type. This finding holds important implications for training methodology, talent identification, and athlete development in this sport. Furthermore, variations in fat mass observed between different subgroups raise new questions, potentially linked to training regimens, selection policies, or regional disparities—warranting further investigation.

## 5. Conclusions

This systematic review provides information that may be useful to technical and medical staff who develop sports programs with blind 5-a-side football players, thus facilitating an adequate assessment of BC and anthropometric factors. The results of this review reveal a tendency toward defining BC as influencing the athletic performance of blind 5-a-side football players. However, it is not conclusive that these improvements occur in response to each playing position. Further studies are needed to analyze the effects of BC on athletic performance, especially when BC is related to other physical, nutritional, technical, and tactical variables during training and competition.

## Figures and Tables

**Figure 1 jfmk-10-00238-f001:**
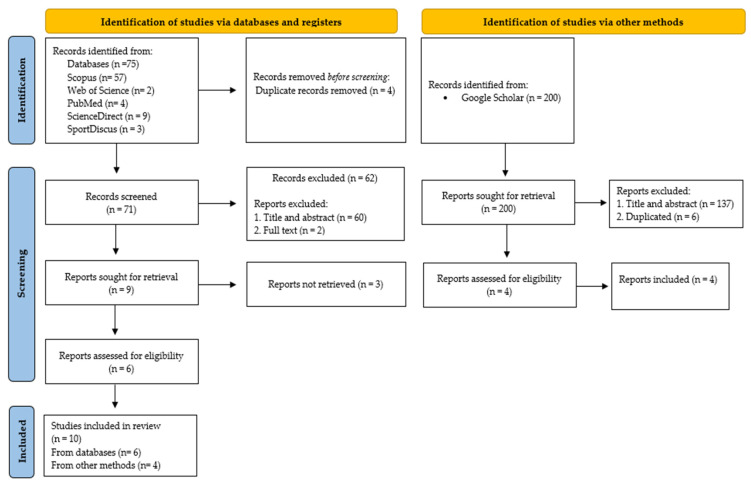
Flow diagram of the systematic review.

**Table 1 jfmk-10-00238-t001:** Inclusion and exclusion criteria.

Population	Intervention	Comparison	Outcomes
Blind 5-a-side players aiming to train or improve their performance	Anthropometry Bioelectrical impedance (BIA) Dual-energy X-ray absorptiometry (DXA)	Measurement methods Equations Performance levels Blind players	Anthropometric characteristics (skinfolds, circumferences, diameters, lengths), somatotype, body composition, fat mass, fat-free mass, muscle mass

**Table 2 jfmk-10-00238-t002:** Methodological quality of the studies evaluated with the PEDro scale.

	Items												
Studies (Author(s)–Year)	1	2	3	4	5	6	7	8	9	10	11	Total PEDro	
Hernández-Beltrán et al. [41]	1	0	0	1	0	0	0	1	1	1	1	6	Good
Lameira Oliveira et al. [40]	1	0	0	1	0	0	0	1	1	0	1	5	Moderate
Esatbeyoglu and Kin-İsler [39]	1	0	0	1	0	0	0	1	1	1	1	6	Good
Sancio et al. [38]	1	0	0	1	0	0	0	1	1	1	1	6	Good
Lameira Oliveira et al. [37]	1	0	0	1	0	0	0	1	1	0	1	5	Moderate
Lameira De Oliveira et al. [36]	1	0	0	1	0	0	0	1	1	0	1	5	Moderate
Lameira De Oliveira et al. [35]	1	0	0	1	0	0	0	1	1	1	1	6	Good
Gorla et al. [34]	1	0	0	1	0	0	0	1	1	1	1	6	Good
Durán-Agüero et al. [33]	1	0	0	1	0	0	0	1	1	1	1	6	Good
Castelli Correia de Campos et al. [32]	1	1	0	1	1	1	0	1	1	0	1	8	Good

**Table 3 jfmk-10-00238-t003:** Classification of the general variables of the selected studies.

Author(s)–Year	N° Part	S	Age (Yrs) and Func Class	Weight (kg)	Height (cm)	IMC (kg/m^2^)	BF (%)	MM (%) * and LM ** (kg)	Clas
Hernández-Beltrán et al. [41]	12	♂	28.7 ± 8.8	73.8 ± 10.7	176.8 ± 9.0	23.57	12.55		
Lameira Oliveira et al. [40]	63	♂	28.0 ± 5.8 (B1)	74.8	170.0	24.9	19.3	80.7 **	Mes-End
Esatbeyoglu and Kin-İsler [39]	12	♂	23.2 ± 3.7 (B1)	79.8 ± 10.9	181 ± 0.08	24.3 ± 2.1	10.53 ± 3.6		
Sancio et al. [38]	8	♂	26.8 ± 6.5 (B1)	81.8 ± 15.7	170.3 ± 5.02		28.12 ± 6.6	43.63 ± 4.5 *	Mes-End
Lameira Oliveira et al. [37]	5	♂	32.6 ± 8.0 (B1)	70.9 ± 10.5	169 ± 7.7	25.1 ± 5.4	20.4 ± 5.1	39.5 ± 3.5 **	
Lameira De Oliveira et al. [36]	13	♂	27.0 ± 6.5 (B1)	71.7 ± 7.4	172.0 ± 6.1	24.1 ± 1.7	15.9 ± 2.9	43.6 ± 2.5 *	Mes-End
Lameira De Oliveira et al. [35]	15	♂	24 ± 5.6 (B1)	71.7 ± 7.4	172 ± 6.1	24.1 ± 1.7	15.9 ± 2.9		Mes-End
Gorla et al. [34]	23	♂	22.5 ± 31 (B1)	64.9–77.9	169–175	22.3–26	10.4–15.9		End-Mes
Durán-Agüero et al. [33]	11	♂	26.4 ± 9.8	71.4 ± 18.9	163.6 ± 16.0	25.1	25.8	45.6 *	Mes
Castelli et al. [32]	6	♂	27.3 ± 5.5 (B1)		1.72 ± 0.09	25.6 ± 1.3	15.9 ± 4.54		End

Note—Ref: Reference; N° Part: Number of participants; Man: S: sex: ♂; Yrs: years; kg: kilograms; cm: centimeters; m: meters; %: percentage; Clas: Classification; Mes: Mesomorph; End: Endomorf; B1: Paralympic classification based on medical criteria for sport for blind people; func class: functional classification; BF: body fat; MM: muscle mass; LM: lean mass. * Evaluation of lean mass in kg; ** Evaluation of muscle mass in %.

**Table 4 jfmk-10-00238-t004:** Classification of the methodological procedures of the studies.

Author(s)/Year	Study’s Aim	Variables	Instruments	Determination of % Fat and Somatotype	Results	Conclusions
Hernández-Beltrán et al. [41]	Analyze BC based on laterality and playing position of the players of the Spanish FpC	National Team PP, laterality, weight, FM, FFM, body water, BMD, AEC/AET, trunk weight, left and right arm weight, left and right leg weight	Tanita BC-601 BC monitor (Tokyo, Japan), SECA wall-mounted height rod (Hamburg, Germany).	NA	Laterality does not differ in playing position in blind 5-a-side players. BC was found to differ in response to playing positions, so determining it can be key when selecting players for a specific position.	BC influences players’ performance and, in turn, is associated with improved health. Low levels of MM increase the likelihood of injury. Therefore, determining players’ BC will allow for the development of specific training sessions aimed at increasing muscle strength.
Lameira Oliveira et al. [40]	To compare BC and somatotype of high-performance blind 5-a-side athletes from different playing positions	PP, skinfolds, body perimeters, bone diameters, height and BW, somatotype	Cescorf caliper (Porto Alegre, Brazil), precision 0.1 mm, Cardiomed pachymeter (Brasília, Brazil), precision 0.1 cm, Sanny Medical flexible metal tape (São Paulo, Brazil), precision 0.1 cm, Soehnle scale (Backnang, Germany), precision 0.1 kg, Soehnle stadiometer (Backnang, Germany), precision 0.1 cm.	Siri formula for body fat percentage [54], Heath–Carter method [55] for somatotype	Wing players presented lower values in body fat percentage (%F = 17.4%) compared to the Closer (23.1%) and Center (21.5%) positions (*p* < 0.05). There is a predominance of the muscular component and a meso-endomorphic somatotype profile overall and for each of the playing positions evaluated.	PP in blind 5-a-side football expresses various variations linked to the specific physical demands of the sport, where BC has been shown to vary in response to playing position. Information on the overall somatotype profile and by PP in blind 5-a-side football can support the development of specific training processes.
Esatbeyoglu and Kin-İsler [39]	To determine sex differences in variables related to BP, BMI, BC, and postural balance in athletes with VI	Balance, BP level, BC, % fat, FM, FFM, BW	International Physical Activity Questionnaire short version, Modified Sensory Integration and Balance Clinical Test Tool, Tanita TBF401A scale (Tokio, Japan), accuracy 0.1 kg, Holtain wall stadiometer (Crosswell, UK).	NA	No statistically significant differences were reported in BC indicators, especially in FM and FFM when comparing sighted athletes with athletes with VI.	Male athletes with VI expressed a higher BMI than women. The BP level demonstrates that VI is not a barrier to maintaining optimal BP levels, while there is a difficulty in balance, even if their PF levels are acceptable. It is suggested to incorporate balance into training sessions in this population group.
Sancio et al. [38]	To analyze the anthropometric profile and its relationship with ball transfer speed in players of the Argentine National futsal Team for the Blind	Weight, height, length of lower limbs, % adipose fat, skinfolds, % MM, muscle adipose ratio, skeletal index, somatotype	Omron^®^ scale, model HBF500INT (Kyoto, Japan), accuracy 0.1 kg; wall-mounted acrylic stadiometer, brand Calibres Argentinos (Rosario, Argentina); Harpenden skinfold caliper, accuracy 0.2 mm; Calibres Argentinos metallic anthropometric tape (Rosario, Argentina)	Heath–Carter method [55] for somatotype	The results express that there is a high correlation between transfer speed and skeletal index (r 0.85 *p* < 0.01), and a moderate correlation with the length of lower limbs (r 0.69) and with variables related to muscle tissue, especially with mesomorphism (r 0.59), kg MM (r 0.57), thigh muscle area (r 0.56) and calf (r 0.55).	Ball transfer speed is related to the anthropometric profile, primarily by the length of the lower limbs and their relationship to trunk length. This allows coaches to consider these variables within the process of selecting and developing players in different short- and long-term sporting processes.
Lameira Oliveira et al. [37]	To describe the anthropometric characteristics and aerobic fitness of blind 5-a-side football players	Skinfolds, bone diameters, body circumferences, BW, height, BMI, % FM, % LM, % bone mass	Cescorf caliper (Porto Alegre, Brazil), accuracy 0.1 mm; Cardiomed pachymeter (Brasília, Brazil), accuracy 0.1 cm; Sanny Medical flexible metal tape (São Paulo, Brazil), accuracy 0.1 cm; Soehnle digital scale (Backnang, Germany), accuracy 0.1 kg; Soehnle stadiometer (Backnang, Germany), accuracy 0.1 cm.	Siri formula for fat percentage [54], Rocha modified von Doblen equation for bone mass [56]	Anthropometric characteristics were consistent with the specificities of the sport and sport level. Likewise, body fat percentage (%F = 20.4) and average VO2 max value. (36.3 ± 4.7 mL^−1^ kg^−1^ min) are lower than those reported by elite athletes in blind 5-a-side football.	Anthropometric characteristics and aerobic fitness are crucial in blind 5-a-side football, as high levels of MM and good aerobic fitness allow athletes to adapt to the physical, technical, and tactical demands of the game. This, in turn, helps improve training processes and athletic performance.
Lameira De Oliveira et al. [36]	To analyze the dermatoglyphic characteristics and BC of blind 5-a-side football players belonging to the Brazilian National Team	Fingerprints, skinfolds, bone diameters, body perimeters, BW, height, BMI, % BF, % LM, % bone mass	Cescorf caliper (Porto Alegre, Brazil), precision 0.1 mm, Cardiomed pachymeter (Brasília, Brazil), precision 0.1 cm, Soehnle digital scale (Backnang, Germany), precision 0.1 kg.	Siri formula for fat percentage [54], Jackson & Pollock equation for BD [57], Heath–Carter method [55] for somatotype	There is a proximity in the reported values in BC between goalkeepers and full-backs with a meso-endomorph profile and between defenders and pivots characterized by a balanced mesomorph profile.	The somatotypic profile of blind 5-a-side football players leans toward the independent muscle component of the PP that they occupy, and this, in turn, is related to the dermatoglyphic characteristics of speed and strength. These genetic and morphological relationships are key to identifying and supporting preparation processes in response to the specific demands of the sport.
Lameira De Oliveira et al. [35]	To analyze the BC and somatotype of the Brazilian Paralympic futsal team athletes at Rio 2016 in response to the playing position	Somatotype, BC, % FM, BD	Cescorf caliper (Porto Alegre, Brazil), precision 0.1 mm, Cardiomed pachymeter (Brasília, Brazil), precision 0.1 cm, Soehnle digital scale (Backnang, Germany), precision 0.1 kg, Soehnle stadiometer (Backnang, Germany), precision 0.1 cm.	Siri formula for fat percentage [54], Jackson & Pollock equation for BD [57], Heath–Carter method [55] for somatotype	The study did not report statistically significant differences in any of the anthropometric variables or BC. Regarding the somatotypic profile, the group was classified as meso-endomorph. The defenders (2.6-4.4-2.4) and the pivots (2.2-5.6-2.3) had a balanced mesomorphic profile, while the goalkeepers (3.2-5.8-1.6) and wings (3.2-5.7-1.6) presented a meso-endomorphic profile.	The team was characterized by its homogeneity in terms of anthropometry and BC, where no differences were reported in response to the playing position. Blind 5-a-side football players present a predominance of the muscular component in the somatotypic profile at a general level and in each of the playing positions.
Gorla et al. [34]	To determine the somatotypic profiles and BC of the Brazilian national blind 5-a-side football team players	BMI, % FM, somatotype, skinfolds, bone diameters, BD	Bascula Plena Acqua^®^ model, WCS wall-mounted stadiometer, Harpenden caliper (Crosswell, UK), precision 0.2 mm, Cardiomed pachymeter (Brasília, Brazil), precision 0.1 cm.	Siri formula for body fat percentage [54], Heath-Carter method [55] for somatotype	Goalkeepers express a statistically significant difference (*p* ≤ 0.05) in the anthropometric variable of CM (82.3 kg) and in the BC variables: %GC (21.5%) and ∑9DC (169.5) compared to the other positions. Regarding the somatotypic profile, there were no statistically significant differences (*p* ≤ 0.05). However, there is a trend toward an endo-mesomorphic profile.	There is a difference in the somatotype of the GK compared to other PPs, which leads to defining the specific training characteristics that each player must receive to respond to the specific demands of the game. The tendency toward an endo-mesomorphic somatotype does not favor high-intensity actions, so it is necessary to focus on reducing FM.
Durán-Agüero et al. [33]	To determine the anthropometric profile of elite Chilean Paralympic athletes by means of BC and somatotype	BC, somatotype	Scale Scale-tronix somatotype (Batesville, USA), precision 0.1 kg, SECA stadiometer (Hamburg, Germany), precision 0.1 cm, Rosscraft anthropometer (Minneapolis, USA), precision 0.1 mm, Sanny measuring tape (São Paulo, Brazil), precision 0.1 mm, Harpenden caliper (Crosswell, UK), precision 0.2 mm.	Determination of BC using the Kerr model [58], the pentacompartmental method. and the Heath–Carter method [55] for the somatotype	The athletes express a predominance toward the meso-endomorph somatotype. Likewise, blind 5-a-side football players have a predominance toward the MM component (45.6%) and bone mass (12.1%), as well as low levels of residual mass (11.6%) compared to other Paralympic athletes.	Chilean elite futsal players present a meso-endomorphic somatotype profile with a predominance of the lower limbs and elevated levels of FM compared to other Paralympic athletes (swimming, wheelchair tennis, and powerlifting). It is necessary to develop training programs that improve BC and nutritional habits.
Castelli Correia de Campos et al. [32]	To analyze the effect of 16 weeks of training on physical fitness (PF) and body composition (BC) in blind 5-a-side football athletes on the Brazilian national team	Skinfolds, widths, body density (BD), body fat percentage, somatotype profile, FM, FFM.	Instruments used included the Plena scale Acqua^®^ model, WCS wall stadiometer, Harpenden caliper (Crosswell, UK) with 0.2 mm precision, and Cardiomed pachymeter (Brasília, Brazil) with 0.1 cm precision.	The Siri formula [54] was used to estimate body fat percentage, the Heath–Carter method [55] and Hebbelinck [59] for somatotype, and the Jackson & Pollock equation [57] for BD	No significant differences (*p* ≤ 0.05) were reported between the absolute values of body weight before (77.08 ± 7.73 kg) and after (76.16 ± 8.38 kg) the tests. The same trend was observed in the BC and somatotype of blind 5-a-side football players.	Sixteen weeks of training are effective for improving physical fitness, although they do not appear to be sufficient to produce changes in body composition indicators in blind 5-a-side football players.

Note. BC: Body composition; BD: body density; BMD: bone mineral density; BMI: body mass index; BW: body weight; cm: centimeters; FM: fat mass; FFM: fat-free mass; GK: goalkeeper; kg: kilograms; LM: Lean mass; mm: millimeters; MM: muscle mass; NA: Not applicable; PA: physical activity; PF: physical fitness; PP: playing position; VI: visual impairment.

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
