# Peer review of "Differences in Anthropometric and Body Composition Factors of Blind 5-a-Side Soccer Players in Response to Playing Position: A Systematic Review"

_jfmk, 2025, doi:10.3390/jfmk10030238_

Round 1
Reviewer 1 Report
Comments and Suggestions for Authors
The paper addresses a highly specialized area of sports science: a systematic analysis of body composition in 5-a-side soccer players with blindness. The topic is considered original significant, as it contributes valuable data to the field of Paralympic sports science.
The introduction is well-founded and clearly identifies a research gap and the method is described in detail and with transparency. The strengths of the using references are broad database search (PubMed, Scopus, Web of Science, etc.) and exclusion of low-quality or non-peer-reviewed studies. The limitations are the heterogeneity in methods section among the included studies prevents a meta-analysis, and only one study had a longitudinal design; the rest were cross-sectional study.
The results section, while extensive, is at times repetitive. Needed better organization.
The main conclusions suggest that the somatotype of players tends toward meso-endomorphic, with some variations in muscle mass, fat mass, and body weight depending on playing position. The practical applications are highlighted for coaches and rehabilitation specialists. The links are made between body composition, performance, and injury prevention, while the importance of positional differences is noted, their direct effect on performance is not fully substantiated.
The references are comprehensive and up-to-date, referencing credible journals.
Comments on the Quality of English Language
The paper is well-written, using academic terminology consistently. Minor inconsistencies in position naming (e.g., "fixed player," "pivot," "closer") are present but do not affect overall comprehension.
Finally, the English could be improved to more clearly express the research.
Author Response
Dear Reviewer,
Thank you for reviewing our manuscript. We appreciate it. We have followed your suggestions point by point to improve the manuscript quality, according to our possibilities. The changes have been made in the full text using the red color so that you can see them. Thanks for your time.
Comments 1.
The results section, while extensive, is at times repetitive. Needed better organization.
Response 1. The manuscript was revised and the wording of the results was improved, leaving only the tables.
Comments 2. The paper is well-written, using academic terminology consistently. Minor inconsistencies in position naming (e.g., "fixed player," "pivot," "closer") are present but do not affect overall comprehension.
Response 2.
The manuscript was reviewed and the names of the playing positions were adjusted throughout the document.
Finally, the manuscript was reviewed again to improve the English language expression of the document.
Thank you for your positive feedback on our research. Your valuable suggestions greatly contributed to the improvement of our work.
Reviewer 2 Report
Comments and Suggestions for Authors
The authors have selected a timely and relevant topic, focusing on a segment of sport science that remains underrepresented yet highly significant—the performance characteristics of team sports played by individuals with visual impairments. Specifically, blind 5-a-side football is a complex discipline that imposes multifactorial demands on the athletes, encompassing sensory integration, motor coordination, and psychological adaptability.
According to current literature, the key performance indicators (KPIs) in this sport include auditory-based spatial orientation, proprioception, technical ball control, and a broad range of physical capabilities. Within the domain of physical performance, anthropometric and somatic characteristics are of particular relevance, a point the authors rightly emphasize.
A major scientific contribution of the manuscript is the presentation of the first known systematic review focusing on the anthropometric and body composition profiles of blind football players. The applied methodology is modern and appropriate; the authors employed analytical tools that allow for a structured synthesis of existing research findings. Despite occasional inconsistencies among primary studies, the resulting meta-analytical outcomes offer valuable integrative insights into the physical profiling of blind football athletes.
A particularly noteworthy result is the lack of significant somatotype differentiation across player positions, with a general dominance of the mesomorphic–endomorphic type. This finding holds important implications for training methodology, talent identification, and athlete development in this sport. Furthermore, variations in fat mass observed between different subgroups raise new questions, potentially linked to training regimens, selection policies, or regional disparities—warranting further investigation.
In conclusion, the manuscript reflects a high standard of academic rigor and contributes new and meaningful knowledge to the literature on adaptive sports science. The absence of certain auxiliary data (e.g., training load, match frequency) is not attributable to shortcomings in the review process, but rather to limitations in the available primary sources.
Author Response
Dear Reviewer,
Thank you for reviewing our manuscript. We appreciate it. We have followed your suggestions point by point to improve the manuscript quality, according to our possibilities. The changes have been made in the full text using the red color so that you can see them. Thanks for your time.
Thank you for your positive feedback on our research. Your valuable suggestions greatly contributed to the improvement of our work.
Reviewer 3 Report
Comments and Suggestions for Authors
I would like to begin by thanking you for the opportunity to review your manuscript entitled “Differences in Anthropometric and Body Composition Factors of 5 Blind Soccer Players in Response to Playing Position: A Systematic Review.”
The study is interesting, timely, and addresses a topic of great scientific and practical relevance within the field of adapted sports.
The manuscript presents an appropriate methodological structure and aligns with PRISMA principles. It provides valuable evidence regarding somatotype and body composition profiles based on playing position in blind 5-a-side football, which undoubtedly has important implications for coaches, physical trainers, and health professionals.
However, I would like to offer the following suggestions with the aim of further improving the clarity, transparency, and applicability of the work:
-
Detail in the information sources the search strategies that were considered in relation to the date. It is not clear whether the search was only conducted on March 15, 2025, or if only studies published up to that date were included. Please clarify.
-
Add an explanation as to why only the following databases were consulted: PubMed (Medline), Scopus, Web of Science, Science Direct, Google Scholar, and ResearchGate, and not others such as Springer, Dialnet, or SPORTDiscus.
-
Include the initials of the two authors who analyzed all retrieved articles for duplicate entries.
-
Clarify the criteria used during the review, discrimination, and eligibility process to resolve, through academic discussion, which studies would be included. Please add an explanation that can help readers better understand the methodology.
-
It would be necessary to add a section on practical applications to help interpret the results of this review more effectively.
Author Response
Dear Reviewer,
Thank you for reviewing our manuscript. We appreciate it. We have followed your suggestions point by point to improve the manuscript quality, according to our possibilities. The changes have been made in the full text using the red color so that you can see them. Thanks for your time.
Reviewer 1 (R1)
Authors (A)
R1
Detail in the information sources the search strategies that were considered in relation to the date. It is not clear whether the search was only conducted on March 15, 2025, or if only studies published up to that date were included. Please clarify.
- We thank the reviewer for his valuable contribution. We have made the suggested change.
All studies published up to March 15, 2025 were retrieved.
R1
Add an explanation as to why only the following databases were consulted: PubMed (Medline), Scopus, Web of Science, Science Direct, Google Scholar, and ResearchGate, and not others such as Springer, Dialnet, or SPORTDiscus.
- We thank the reviewer for his valuable contribution. We have made the suggested change.
These databases were consulted for use in various reviews and were used to search data-bases and other sources. And, to clarify, if studies were searched for in SPORTDiscus
R1
Include the initials of the two authors who analyzed all retrieved articles for duplicate entries.
- We thank the reviewer for his valuable contribution. We have made the suggested change.
Two authors searched independently (B.A.B.-P. and J.O.-A.). The purpose was to identify papers that met the criteria (Table 1).
R1
Clarify the criteria used during the review, discrimination, and eligibility process to resolve, through academic discussion, which studies would be included. Please add an explanation that can help readers better understand the methodology.
- We thank the reviewer for his valuable contribution. We have made the suggested change.
The academic debates for the inclusion of the studies took into account the duplicate search by two authors on two different days to review the documents. In particular, the methodology (study design, variables, instruments, determination of fat percentage, and somatotype) was reviewed, as well as the results and main conclusions (Table 4).
R1
It would be necessary to add a section on practical applications to help interpret the results of this review more effectively.
- We thank the reviewer for his valuable contribution. It has been corrected.
4.3.3. Practical applications
A particularly noteworthy result is the lack of significant somatotype differentiation across player positions, with a general dominance of the mesomorphic–endomorphic ty-pe. This finding holds important implications for training methodology, talent identifica-tion, and athlete development in this sport. Furthermore, variations in fat mass observed between different subgroups raise new questions, potentially linked to training regimens, selection policies, or regional disparities—warranting further investigation.
Thank you for your positive feedback on our research. Your valuable suggestions greatly contributed to the improvement of our work.
Best regards
Reviewer 4 Report
Comments and Suggestions for Authors
Overall: The purpose of this study was to systematically synthesize the existing evidence on differences in anthropometric characteristics and body composition among blind 5-a-side football players according to playing position, and to derive practical recommendations for researchers and coaches. The authors completed a thorough search of the scientific literature and constructed a very useful literature review table for researchers in this line of inquiry. However, presentation of some of the results in a compiled fashion would be beneficial, along with enhancement of the practical applications to coaches discussion.
Table 2: Consider moving Table 2 (and Lines 155-160) to the Results as it's really a description of the results of the literature review.
Lines 163-185: All that is reported in the Results text is a basic description of the number of studies included, with all other information included in Figure 1 and Tables 3 and 4. It would help the reader if some basic descriptive information was included in the text to summarize some of the key things that are subsequently discussed in the Discussion (see below).
Lines 190-201: This is really the first presentation of these overall results (vs. a description of each specific study in a large table). Please consider moving this information to the Results so that the rest of the Discussion has better context. In addition, please consider expanding the description a bit further by including some compiled descriptive statistics to support these statements. For example, if the somatotype of blind 5-a-side football players tends toward meso-endomorphic, provide a compiled frequency description of these somatotypes across the papers. These kinds of descriptive statistics would really help support the points subsequently made in the Discussion.
Lines: 314-322: Although some recommendations for future research is integrated throughout the Discussion, this is really the only section that provides practical recommendations for coaches. Consider taking some of these sentences and expanding them into a new section in the Discussion that is more specific to coaches, especially since this was one of the stated aims of the study (Lines 87-88). The Conclusions section could then be more broad in nature and summative of all the study aims.
Author Response
Dear Reviewer,
Thank you for reviewing our manuscript. We appreciate it. We have followed your suggestions point by point to improve the manuscript quality, according to our possibilities. The changes have been made in the full text using the red color so that you can see them. Thanks for your time.
Comments 1. Table 2: Consider moving Table 2 (and Lines 155-160) to the Results as it's really a description of the results of the literature review.
Response 1. The changes suggested by the reviewer were accepted. Tables 2, 3 and 4 together with figure 1 were moved to socialize the results section. Likewise, the review report was added to the INPLASY platform with its respective code.
Comments 2. Lines 163-185: All that is reported in the Results text is a basic description of the number of studies included, with all other information included in Figure 1 and Tables 3 and 4. It would help the reader if some basic descriptive information was included in the text to summarize some of the key things that are subsequently discussed in the Discussion (see below).
Response 2.
The changes suggested by the reviewer were accepted. A brief description was included before each table to improve the interpretation of the results.
The sections are as follows:
3.3. Analysis of the participants
The 10 studies comprising the sample of this systematic review included 168 athletes, all of whom were men. Table 3 specifies the characteristics of the sample selected.
3.4. Analysis of the studies
Table 4 specifies the characteristics of the sample selected for each study (study aim, variables, results, instruments, determination of % fat and somatotype, conclusions).
Comments 3. Lines 190-201: This is really the first presentation of these overall results (vs. a description of each specific study in a large table). Please consider moving this information to the Results so that the rest of the Discussion has better context. In addition, please consider expanding the description a bit further by including some compiled descriptive statistics to support these statements. For example, if the somatotype of blind 5-a-side football players tends toward meso-endomorphic, provide a compiled frequency description of these somatotypes across the papers. These kinds of descriptive statistics would really help support the points subsequently made in the Discussion.
Response 3. Many thanks to the reviewer for his valuable suggestions.
The information that is socialized in the different sections of the discussion, effectively evidences the results that all the studies allow. In table 2 where the somatotypes of the athletes are shared, there are two studies that do not report it, while there are 4 studies that report that the prevailing somatotype is the meso-endomorphic.
The somatotype of blind 5-a-side football players tends toward meso-endomorphic [35,36,38,40].
Comments 4. Lines: 314-322: Although some recommendations for future research is integrated throughout the Discussion, this is really the only section that provides practical recommendations for coaches. Consider taking some of these sentences and expanding them into a new section in the Discussion that is more specific to coaches, especially since this was one of the stated aims of the study (Lines 87-88). The Conclusions section could then be more broad in nature and summative of all the study aims.
Response 4. These sections have been restructured to improve the practical applications of the study. Three sections were designed: 1) limitations and strengths; 2) future recommendations; and 3) practical applications.
4.3.1. Limitations and Strengths
This study has several limitations, which are outlined below. There is diversity in the number of participants evaluated; the most commonly used study designs have been cross-sectional, with only one longitudinal study reported. Another limitation is that not all studies include goalkeepers, making it difficult to generate a deep understanding of BC in response to playing positions. Similarly, there is diversity in the variables that determine anthropometric and BC factors. This prevented comparative measurements for each of the variables analyzed in the studies included in this review. The studies included in this systematic review were too heterogeneous and of moderate methodological quality, making it impossible to conduct a meta-analysis. Although this type of study, which was conducted to analyze blind 5-a-side football, does not allow for solid conclusions, the information contained in Table 4 reflects important information from each study that could be further explored by the scientific community.
4.3.2. Future recommendations
Future research directions for studying BC in blind 5-a-side football should seek associations with other performance indicators, especially nutritional, physical, and external load variables in competitions. Likewise, longitudinal studies, randomized controlled trials are related or principal component analyses based on standardized and recognized protocols are needed to understand how different variables are related to playing position. Furthermore, the findings of this study should be carefully analyzed to be incorporated into different training processes and the practical work of coaches and athletes. Finally, it would be important to conduct BC assessments in blind female 5-a-side football players, as no studies analyzing these factors in this population sample have been reported. This will allow us to further expand the horizons of this sport. Because it is still a practice that has low scientific productivity and requires an increasing amount of research [65].
4.3.3. Practical applications
A particularly noteworthy result is the lack of significant somatotype differentiation across player positions, with a general dominance of the mesomorphic–endomorphic type. This finding holds important implications for training methodology, talent identification, and athlete development in this sport. Furthermore, variations in fat mass observed between different subgroups raise new questions, potentially linked to training regimens, selection policies, or regional disparities—warranting further investigation.
Thank you for your positive feedback on our research. Your valuable suggestions greatly contributed to the improvement of our work.
Round 2
Reviewer 4 Report
Comments and Suggestions for Authors
The authors have addressed previous reviewer comments.